# Intestinal-Failure-Associated Liver Disease: Beyond Parenteral Nutrition

**DOI:** 10.3390/biom15030388

**Published:** 2025-03-08

**Authors:** Irene Mignini, Giulia Piccirilli, Federica Di Vincenzo, Carlo Covello, Marco Pizzoferrato, Giorgio Esposto, Linda Galasso, Raffaele Borriello, Maurizio Gabrielli, Maria Elena Ainora, Antonio Gasbarrini, Maria Assunta Zocco

**Affiliations:** CEMAD Digestive Diseases Center, Fondazione Policlinico Universitario “A. Gemelli” IRCCS, Università Cattolica del Sacro Cuore, Largo A. Gemelli 8, 00168 Rome, Italy; irene.mignini@guest.policlinicogemelli.it (I.M.); giulia.piccirilli01@icatt.it (G.P.); federica.divincenzo01@icatt.it (F.D.V.); carlo.covello01@icatt.it (C.C.); marco.pizzoferrato@policlinicogemelli.it (M.P.); giorgio.esposto@guest.policlinicogemelli.it (G.E.); linda.galasso@guest.policlinicogemelli.it (L.G.); raffaele.borriello@unicatt.it (R.B.); maurizio.gabrielli@policlinicogemelli.it (M.G.); mariaelena.ainora@policlinicogemelli.it (M.E.A.); antonio.gasbarrini@unicatt.it (A.G.)

**Keywords:** Short bowel syndrome, intestinal failure, intestinal-failure-associated liver disease (IFALD), parenteral nutrition

## Abstract

Short bowel syndrome (SBS), usually resulting from massive small bowel resections or congenital defects, may lead to intestinal failure (IF), requiring intravenous fluids and parenteral nutrition to preserve patients’ nutritional status. Approximately 15% to 40% of subjects with SBS and IF develop chronic hepatic damage during their life, a condition referred to as intestinal-failure-associated liver disease (IFALD), which ranges from steatosis to fibrosis or end-stage liver disease. Parenteral nutrition has been largely pointed out as the main pathogenetic factor for IFALD. However, other elements, such as inflammation, bile acid metabolism, bacterial overgrowth and gut dysbiosis also contribute to the development of liver damage and may deserve specific treatment strategies. Indeed, in our review, we aim to explore IFALD pathogenesis beyond parenteral nutrition. By critically analyzing recent literature, we seek to delve with molecular mechanisms and metabolic pathways underlying liver damage in such a complex set of patients.

## 1. Introduction

Short bowel syndrome (SBS) is a rare condition resulting from extensive resection of the small intestine, which leads to a significant decrease in the capacity for intestinal absorption. The classic anatomical definition of SBS is based on a residual length of the post-duodenal small bowel measuring less than 200 cm. Although this threshold is seldom used in clinical practice, it indicates a loss of more than half of the normal length of the small bowel [1]. The consequent intestinal failure (IF) is marked by severe malabsorption, necessitating intravenous nutritional supplementation (IVS) to maintain homeostasis and/or support growth [2].

The exact incidence of SBS is not thoroughly documented; however, current estimates suggest that approximately 5 to 10 new cases occur annually per 1,000,000 individuals, with a smaller proportion of these cases which may require home parenteral nutrition (HPN) [3]. SBS primarily arises from ischemic events, responsible for approximately 45% of cases attributed to mesenteric arterial or venous ischemia. Chronic enteropathy, including post-radiation enteritis and refractory sprue, accounts for around 25% of cases. Crohn’s disease contributes to 5–10% of instances, while surgical complications are implicated in about 10%. The remaining cases stem from volvulus, traumatic injuries, and various other rare etiologies [3].

SBS is classified according to anatomical assembly into 3 types: patients with terminal enteroanastomosis (SBS-J or type I); patients with jejuno-colic anastomosis (SBS-JC or type II); and those with jejunoileal anastomosis, preserving the entire colon and ileocecal valve (SBS-JIC or type III) [4]. This classification not only holds anatomical significance, but it also affects the level of dependency on parenteral nutrition (PN) and, in turn, the likelihood of developing IF along with the patients’ prognosis [4,5]. The variability in prognosis and dependence on PN is influenced by intestinal adaptability, particularly by the colon, which enhances absorption, as well as the ileocecal valve and terminal ileum, which regulate the flow of chyme [6,7,8]. The length of the residual small intestine also plays a crucial role weaning off PN, especially when the residual length below the Treitz ligament exceeds 75 cm [3].

SBS is a significant contributor to IF, characterized by pronounced malabsorption of macronutrients, fluids, and electrolytes, necessitating IVS to uphold patient survival and promote growth. Chronic intestinal failure (CIF) manifests when compromised bowel function endures for an extended duration—typically months or years—in metabolically stable individuals. These patients can be effectively managed through HPN programs, which may lead to reversible or irreversible outcomes, contingent upon the potential for recovery of residual bowel function [9]. The management of IF goes beyond merely providing PN; it requires a comprehensive approach that combines nutritional and pharmacological interventions. Strategies to promote intestinal adaptation include maintaining an oral diet when possible, performing reconstructive surgeries, and using trophic agents like the glucagon-like peptide-2 (GLP-2) analogues, which help enhance the growth and function of the remaining intestinal lumen [10]. The goal is to reduce reliance on PN and alleviate PN-related complications, including electrolyte imbalances, infections, thrombosis, bone disease, metabolic disturbances and liver damage [11,12]. Indeed, liver impairment, particularly intrahepatic cholestasis, was observed following the introduction of PN into clinical practice [13]. This condition was originally referred to as parenteral nutrition-associated liver disease (PNALD). However, multiple lines of evidence indicates that the deterioration of liver function in patients undergoing long-term HPN is not solely attributable to PN itself but rather to the underlying IF and its associated complications [11]. As a result, the term PNALD has been redefined as IF-associated liver disease (IFALD). The European Society for Clinical Nutrition and Metabolism (ESPEN) states that “the term IFALD refers to liver injury as a result of one or more factors relating to IF including, but not limited to, PN and occurring in the absence of another primary parenchymal liver pathology (e.g., viral or autoimmune hepatitis), other hepatotoxic factors (e.g., alcohol/medication) or biliary obstruction” [9,14].

Currently, our understanding of the factors and pathogenic pathways contributing to the development of IFALD remains limited. IFALD is a complex and multifactorial condition involving several interrelated elements, including those associated with intestinal insufficiency, dysregulation of the gut–liver axis, and factors arising from the administration of PN [11]. Among PN-related factors, macronutrient overload and the use of soy-based lipid emulsions (SO-ILE) promote liver fat deposition, inflammation, and liver damage. Micronutrient deficiencies and bloodstream infections related to central venous catheters (CRBSIs) further exacerbate liver damage and dysfunction [15,16,17]. Other pathogenic factors contributing to IFALD include mucosal atrophy, alterations in the microbiome, and increased intestinal permeability, which allow bacterial lipopolysaccharide (LPS) to reach the liver, causing inflammation and liver damage [18,19]. Malabsorption of bile acids and dysfunction of farnesoid-X receptor (FXR)/fibroblast growth factor-19 (FGF19) axis worsen cholestasis and liver fibrosis [18].

This review thoughtfully examines the pathogenesis of IFALD, expanding the focus beyond the traditional emphasis on PN. By thoroughly exploring recent literature, we aim to enhance our understanding of the molecular mechanisms and metabolic pathways that contribute to liver injury. Additionally, we will investigate emerging therapeutic approaches and future perspectives, with a particular emphasis on gut microbiota modulation, regulation of bile acid metabolism, optimization of nutritional formulations, and the development of innovative surgical and pharmacological treatments.

## 2. Clinical Presentation and Diagnosis of IFALD

According to the ESPEN position paper, the diagnosis of IFALD should be based on biochemical alterations of liver function tests (LFTs) and/or evidence of radiological and/or histological liver abnormalities arising in an individual with IF, after excluding other primary parenchymal liver pathology such as ischemic, viral or autoimmune hepatitis, hepatotoxic factors (e.g., alcohol/medications) or biliary obstruction. Liver histology is not considered mandatory and should be evaluated on a case-by-case basis [11,14]. Even though no formal agreement has been reached so far, diagnostic criteria for IFALD are represented by the elevation of cholestatic liver enzymes, notably alkaline phosphatase (ALP) and γ-glutamyl transferase (γGT) > 1.5 times above the upper limit of normal (ULN) that persists for >6 months in adults, or >6 weeks in children [17,20]. Furthermore, IFALD causes the increase of alanine aminotransferase (ALT) > 2–3 ULN and total bilirubin 2–3 times the ULN. ALP is predominantly elevated in the majority of patients with IFALD [21]. Another important assessment is the temporal association between PN administration and elevation of LFTs [22]. The elevation of liver enzymes usually occurs between the first and third week after the initiation of PN [23]. In patients with acute intestinal failure (AIF), LFTs are often mildly elevated and related to the underlying metabolic impairment, which reverts as soon as the metabolic state becomes stable and an enteral or oral diet is reestablished, despite continuing PN [9]. In patients with chronic CIF, alterations of liver function may evolve to liver failure, a mandatory indication for a life-saving combined liver and small bowel transplantation [9,11].

IFALD is associated with a wide spectrum of clinical manifestations, from mildly impaired LFTs to severe cholestasis, steatosis/steatohepatitis, and fibrosis. The progression to cirrhosis, end-stage liver failure, is rare in adults compared to children [22,24]. Infants are more prone to PN-related hepatic injury and develop fibrosis more rapidly than adults [25].

Regular monitoring of IFALD progression is essential to avoiding its advancement to end-stage liver disease and to determining the appropriate timing for patient referral for liver and small bowel transplantation. Routine evaluation of LFTs every 3 to 4 months is recommended [14]. To date, multiple studies have employed different diagnostic approaches to assess IFALD incidence and outcomes. Due to the absence of a standardized diagnostic criteria for IFALD, a wide range of prevalence (4.3% to 65%) has been reported, depending on the diagnostic criteria used [26,27,28,29]. For instance, Cavicchi et al. identified chronic cholestasis in HPN-dependent adults as the presence of LFTs (γGT, ALP or serum conjugated bilirubin) at 1.5 times the ULN in at least two of three markers, persisting for a minimum of 6 months [30]. Later, Luman and colleagues defined IFALD by detecting any liver function parameter exceeding 1.5 times the normal range at least 6 months after initiating HPN. They noted that elevated ALP was the most frequent alteration, appearing in 39% of patients, though none of the patients developed decompensated liver disease during extended follow-up [31]. Other blood-based parameters validated in different liver conditions—such as aspartate aminotransferase (AST) to ALT ratio (AAR) [32], AST to platelet ratio index (APRI) [33], and the fibrosis index based on the 4 factor (FIB-4) index [34]—may have limitations for IFALD diagnosis. The unique pathophysiology involving intestinal failure, chronic parenteral nutrition, and sepsis-related liver injury, which differs from the other liver diseases in which these indices were originally validated, reduces their accuracy in this specific context; extrahepatic influences that can affect these biomarkers independently of liver fibrosis; the lack of direct validation against liver histology; diverse liver injury patterns, such as cholestasis, steatosis, and fibrosis; the impact of chronic inflammation due to parenteral nutrition, infections and metabolic alterations. A recent single-center study systematically demonstrates the variation on prevalence and incidence of IFALD depending on diagnostic criteria used, confirming the need for a consensus definition to be used between different national and international IF units [35].

## 3. PN and Its Role in IFALD Pathogenesis

Over the last 50 years, PN has become a standard of care in SBS treatment, with a variety of formulations now available for both in- and out-patients to ensure optimal management and individualized care [36]. PN formulation consists of proteins (amino acids), carbohydrates (dextrose), lipids, electrolytes, minerals, vitamins, trace elements, and water. It can be prepared as a “2-in-1” solution, where the intravenous fat emulsion (IVFE) is administered separately from the dextrose and amino acids, or as a total nutrient admixture (TNA or “3-in-1”), which combines all components, including fat, in a single bag [37]. Despite its crucial role, PN imposes significant healthcare costs and negatively affects patients’ quality of life, disrupting daily activities, sleep patterns, and sociality [2,38,39]. Additionally, it remains associated with a considerable risk of complications, including infections, catheter-related issues, thrombosis, metabolic imbalances such as metabolic bone disease, iron-deficiency anemia and manganese toxicity, kidney and liver disorders, and ultimately kidney and liver failure. Indeed, in the pathogenesis of IFALD, PN plays a major role, contributing to liver injury due to both macronutrient excess and micronutrient deficiencies [40].

Patients receiving PN have a lack of micronutrients like carnitine, taurine, and choline. In particular, carnitine, an essential co-factor of mitochondrial β-oxidation involved in the regulation of liver regeneration [41], is as low as 50% of the normal values [42]. L-carnitine supplementation has been associated with prevention of non-alcoholic steatohepatitis progression in a mouse model [43], while in humans it proved to prevent muscle loss in cirrhotic patients [44] and to improve hepatic steatosis in patients with nonalcoholic fatty liver disease (NAFLD) and diabetes [45]. However, interventional studies have not demonstrated the benefit of L-carnitine supplementation during PN in patients with abnormal liver tests and low plasma carnitine concentrations, suggesting that carnitine deficiency is not a major cause of IFALD. Similarly, a decrease in plasmatic taurine, a sulfur-containing amino acid, may impact bile acid metabolism by reducing tauro-conjugated bile acid formation [46,47]. Animal studies showed that taurine supplementation enhances bile flow and prevents cholestasis [48]. In neonatal patients, taurine appears to provide protection against PN-associated cholestasis [49]. However, a clinical study in postsurgical adult patients receiving short-term PN (5–7 days) found no significant effect of taurine on liver function parameters [50]. Also, choline is crucial for hepatic very low density lipoprotein secretion and lipoprotein homeostasis [51]. Choline deficiency has been found in up to 80% of long-term PN patients and low plasma levels are linked to liver enzyme abnormalities and progressive liver disease [52]. Choline-supplemented PN may contribute to the reversal of these abnormalities [53,54]. Given the disrupted (anti)oxidant balance associated with PN dependency and the role of lipid peroxidation in IFALD development and progression, vitamin E deficiency has been proposed as a possible contributing factor to IFALD. Vitamin E deficiency has been linked to liver steatosis [55,56]. In piglet models, vitamin E supplementation has been shown to prevent increases in biliary and lipidemic markers of IFALD [57]. Lipid emulsions serve as the primary source of vitamin E for patients reliant on PN. As the main lipid-soluble antioxidant, α-tocopherol plays a crucial role in minimizing lipid peroxidation [58]. In PN carbohydrates are represented in the form of hydrous dextrose. Increased dextrose infusion in PN at rates of >5 mg/kg/min induces steatosis whose severity correlates to carbohydrate intake. An experimental animal model with rats demonstrated that animals who received the highest carbohydrate intake (25% dextrose) had the highest liver lipid content and most severe morphological abnormalities, while reducing dextrose concentration to 15% minimized liver fat accumulation [59]. The mechanism involves increased liver lipogenesis due to insulin stimulation [60], and impaired fatty acid oxidation which leads to triglyceride accumulation in hepatocytes [61]. Another crucial mechanism is the elevation of portal venous insulin-glucagon molar ratio, that promotes fatty acid biosynthesis. According to the American Society for Parenteral and Enteral Nutrition guidelines, the recommended glucose infusion rate is 4–5 mg/kg of body weight/min in adults, but it has to be adjusted to each patient in order to achieve normoglycemia [62]. The adverse effects of insulin hypersecretion may also explain why continuous PN infusions exacerbate hepatic dysfunction compared to cyclic infusion. Allowing ≥8 h each day without parenteral glucose infusion has been shown to lower insulin levels and improve AST, ALT, GGT, and bacterial translocation levels, suggesting that cyclic PN may help reverse IFALD. For practical reasons, many long-term PN patients at home follow a nocturnal feeding schedule, making this approach more feasible [63].

IVFEs serve as a source of energy and essential fatty acids, including linoleic acid (a precursor of long-chain ω-6 fatty acids) and α-linolenic acid (a precursor of ω-3 fatty acids). Replacing part of the glucose-derived energy (up to 30%) with parenteral lipids has demonstrated to lower the occurrence of steatosis [64]. However, lipid excess may also increase hepatic disorders. In particular, the adverse effects of lipid administration are related to the type and rate of administration. In all human studies, adverse effects were observed with infusion rates ≥ 110 mg/kg/h or ≥ 1 kcal/kg/h [65]. These findings highlight the importance of avoiding short-duration IVFE infusions. It is generally recommended that IVFEs should provide 20–30% of the total daily caloric intake in PN and should be administered continuously [66].

Currently, most of the commercially available IVFEs in the United States are derived from soybean oil (Intralipid, Nutrilipid). Historically, another type of oil used was safflower oil because of its high ratio of ω-6 fatty acids to ω-3 fatty acids and its high content of phytosterols. However, there are concerns about possible negative effects of PN lipids derived exclusively from vegetable oils, regarding stress response, sepsis, and liver function. Indeed, the administration of SO-ILE and other plant-based-ILEs has been clearly associated with the development of liver damage in adult patients, especially IFALD-cholestasis [30]. It has been hypothesized that the relatively high concentrations of ω-6 polyunsaturated fatty acids (PUFA) in plant-based ILEs generate pro-inflammatory eicosanoids and contribute to hepatic inflammation and cholestasis [24]. To address these concerns, alternative lipid emulsions have been developed, incorporating ingredients such as the anti-inflammatory ω-3 PUFA found in fish oil-based emulsions (FO-ILE), medium-chain triglyceride (MCTs), and olive-oil-based lipid emulsions [67]. Indeed, some studies reported different approaches to obtain the reversal of IFALD-related cholestasis in infants, such as the reduction of the amount of SO-ILEs in PN solutions from 2–3 g/kg/day to 1 g/kg/day [68], the replacement of SO-ILEs with a FO-ILE at 1 g/kg/day [69], or the use of a mixed oil lipid emulsion (MO-ILE) containing fish oil at 2–3 g/kg/day [70]. These findings suggest that either the ω-3 PUFAs in FO-ILEs exert an anti-inflammatory protective effect or that certain components of SO-ILEs, such as ω-6 PUFAs or plant sterols (phytosterols), contribute to IFALD development. A newly available IVFE product, an example of the so-called third-generation combination lipid emulsions, is SMOF lipid (Fresenius Kabi, Bad Homburg, Germany). SMOF lipid consists of a combination of soybean oil, MCTs, olive oil, and fish oil and has an increased ratio of ω-3 PUFA to ω-6 PUFA. Multiple studies suggest that olive oil-based ILE has better liver tolerability than SO-ILE, with lower rates of cholestasis and liver enzyme elevation [71,72]. Muhammed et al. examined in a small group of children the effect on PN-associated jaundice when changing the type of ILE, without reducing the total amount of lipid given [73]. They observed that children with PN-associated jaundice showed improvement when switched from a SO-ILE (Intralipid) to SMOFlipid, with 5 out of 8 patients achieving complete resolution. In contrast, those on Intralipid experienced increased bilirubin levels over six months. Klek et al. carried on a multicentre study in 73 patients with stable IF, requiring PN with SMOFlipid or Intralipid for 4 weeks. They demonstrated that, while within the normal range, LFT values were significantly lower with SMOFlipid compared to Intralipid over 4 weeks [74]. A single-center retrospective study conducted by Jackson et al. evaluated the incidence of PN-associated cholestasis in patients admitted to the neonatal intensive care unit who received either Intralipid 20% or SMOFlipid for ≥14 days, evidencing that using SMOFlipid as the lipid emulsion component of PN may be beneficial in prevention of PNAC in NICU patients [75]. Animal studies using isocaloric and isolipidic diets with varying ω-6 and long-chain ω-3 PUFAs found that mice on an ω-6 diet had higher body and liver weight, total lipid levels, and abdominal fat deposits as well as increased arachidonic acid levels and reduced ω-3 fatty acids compared to those on an ω-3 diet. These findings highlight the role of PUFA composition in hepatic pathology and inflammation [76]. Furthermore, a recent meta-analysis suggests that ω-3 PUFA supplementation can decrease hepatic steatosis and improve liver function parameters in NAFLD [77]. Based on such data, ESPEN recommends limiting SO-ILE administration and reducing the ω-6/ω-3 ratio whenever possible to avoid hepatic steatosis and inflammation [15].

In conclusion, proper regulation of PN components and dosage is crucial for preventing and managing IFALD. Figure 1 schematically represents the imbalance of PN components contributing to liver injury.

## 4. Beyond PN: Other Etiopathogenetic Factors

IFALD pathogenesis is multifactorial: beyond PN-induced mechanisms, it is also influenced by systemic factors, such as the lack of enteral/oral feeding, disruption of enterohepatic bile acid circulation, and changes in the gut microbiome. Indeed, SBS is associated with complex adaptative processes to compensate for the lack of nutrients in the intestinal lumen and/or for the reduced intestinal absorptive function [78]. Such spontaneous responses may unfortunately contribute to liver injury, which is not only related with PN per se but also with the pathophysiological mechanisms of intestinal adaptation, thus justifying why the broader definition of IFALD is currently preferred to the old term PNALD. The worst liver damage occurs in case of total PN, while an even small amount of enteral nutrition is able to maintain the gut-liver signaling mitigating hepatic injury [19].

Bile acids dysmetabolism has been observed in SBS patients, impairing a number of functions, including lipid- and fat-soluble vitamin absorption and lipid and glucose metabolism. In normal enterohepatic circulation, primary bile acids produced by the liver are converted into secondary bile acids through deconjugation by gut microbiota. Since primary bile acids primarily activate FXR and secondary bile acids preferentially target Takeda-G-protein-coupled receptor 5 (TGR5), gut microbes play a key role in regulating bile acid signaling [79,80]. TGR5 drives secretion of crucial metabolic hormones such as gastrin, glucagon, secretin, motilin, pancreatic polypeptide, cholecystokinin and glucagon-like peptide-1 and 2 (GLP-1/2), which respectively increases insulin sensitivity and release and regulates hepatic steatosis and gut growth and integrity [81,82]. In the lack of enteral feeding and nutrients, there is a disruption of the TGR5 signaling pathway, leading to decreased secretion of gut-trophic factors and hormones. Diminished levels of GLP-1 have been associated with steatosis [83]. Changes in bile acid composition disrupt FXR/FGF19 signaling, which is crucial for liver homeostasis. Indeed, under healthy conditions, bile acids mediate intestinal FXR activation, subsequently inducing expression of FGF19 in the gut, which regulates hepatic bile acid synthesis by providing negative feedback control through the inhibition of cytochrome P450 7a1 (CYP7A1) [84,85]. There is evidence that patients with IF have an altered bile acid composition, associated with decreased serum FGF19 levels and increased hepatic CYP7A1 [86]. The subsequent increased hepatic bile acid synthesis may be considered both an adaptative mechanism to enhance lipid absorption and a driver of liver injury leading to increased cholestasis.

Interestingly, in infants with IF, Xiao et al. found that FXR/FGF19 serum levels were inversely correlated with proinflammatory cytokines, such as interleukin 6 and tumor necrosis factor α (TNFα), suggesting that inflammation may worsen liver injury [86]. Recently, Fligor et al. conducted a study on a pre-term piglet model of IFALD to investigate molecular mechanisms underlying early IFALD. In piglets receiving PN, they found many upregulated genes mainly involved in inflammatory signaling, notably transmembrane receptors (TNFα receptors, toll-like receptors—TLRs, interferon receptors) and adhesion molecules. Basing on these results, the authors speculate on a potential therapeutic role of anti-inflammatory agents, including biologic drugs, to prevent IFALD development and progression [87].

In such a context, the role of gut microbiota appears especially intriguing. It is not surprising that extensive bowel resections, the lack of enteral nutrition and the administration of PN contribute to gut dysbiosis, which can be considered part of the intestinal adaptative process. For instance, SBS patients show a predominance of Proteobacteria, especially *Enterobacteriaceae*, which can metabolize broader classes of substrates but are also associated with intestinal and liver inflammation, while anaerobic microorganisms, such as *Clostridiaceae*, are less represented due to the high oxygen levels after bowel resections [18]. In healthy conditions, the lactate produced by gut microbiota is consumed by anaerobic bacteria or converted into other metabolites, like short-chain fatty acids. In SBS, the depletion of anaerobic bacteria contributes to lactate accumulation, increasing the risk of lactate acidosis and encephalopathy, especially in case of impaired liver function [88]. PN may further contribute to dysbiosis: in pre-term neonates, it has been observed that PN, if compared with enteral nutrition, was associated with a reduction of gram-positive and gram-negative bacteria, while a higher number of sepsis and deaths was reported [89].

Interestingly, the specific microbiota composition in SBS patients is interconnected with the other aforementioned pathogenetic mechanisms, defining a fascinating interplay between gut microbiota, PN, bile acid metabolism, and the immune system [18]. Among the bacterial species enriched in SBS microbiota, *Lactobacilli* are worth noting as an example of interconnection between dysbiosis and altered bile acids metabolism. Their role in promoting liver disorders is known from patients affected by NAFLD, whose severity correlates with intestinal bacteria overgrowth [90,91]. Korpela et al. performed microbiota analysis and liver biopsies in a group of patients with pediatric onset IF, finding an increase of *Lactobacilli*, especially *L. plantarum*, which have a high bile salt hydrolase activity, resulting in enhanced bile acid deconjugation, impaired lipid absorption, and altered FXR/FGF19 signaling [92]. In the same study, they confirmed the overabundance of Proteobacteria, which produce LPS, a pro-inflammatory and hepatotoxic compound—they consistently found an association between Proteobacteria, liver steatosis, and intestinal inflammation [92]. Moreover, Proteobacteria express many TLR ligands such as LPS, further exacerbating the pro-inflammatory status [18].

The main etiopathogenetic factors contributing to IFALD development and progression are represented in Figure 2, together with the potential therapeutic strategies, discussed throughout the main text.

## 5. Therapeutic Strategies and Future Perspectives

### 5.1. Bile Acid Signaling Targets

Drugs modulating bile acid signaling pathways are increasingly recognized for their potential to support immune and metabolic homeostasis in the liver. Therapeutic strategies encompass bile acid receptor ligands and bile acid replacement therapy.

#### 5.1.1. Bile Acid Receptor Ligands

Bile acid receptor activation, particularly of FXR, reduces hepatic bile salt load and has been linked to anti-inflammatory effects, positioning FXR as a promising therapeutic target in hepatic diseases. However, its use in IF yields mixed results. In murine models, the FXR agonist GW4064 significantly improved liver histology, reduced serum liver enzymes, and decreased oxidative stress [93]. Similarly, Tropifexor, a FXR agonist, demonstrated efficacy in neonatal piglets, stimulating fibroblast growth factor 19 (FGF19) expression in ileal epithelial cells, increasing portal FGF19 levels, and modulating bile salt hydrolase (BSH) and 7α-dehydrogenation-producing bacteria [94]. Conversely, chenodeoxycholic acid (CDCA), and obeticholic acid (OCA) either had no effect or exacerbated liver injury in SBS piglets [95,96]. Possible explanations include insufficient activation of intestinal FXR due to impaired receptor expression or function in the short bowel. Indeed, FXR activation by bile acids significantly ameliorated IFALD in animal models with an intact gut. Secondarily, the intestinal FXR rather than liver FXR could play a crucial role in the pathogenesis of IFALD. Interestingly, while OCA and CDCA failed to upregulate intestinal FXR and its downstream targets in SBS models, GW4064 successfully enhanced gut FXR signaling and increased FGF15 (murine analog of human FGF19).

Ursolic acid, an agonist of the TGR5, proved to be effective as an oral GLP-2 secretagogue in piglets receiving PN, but it failed to stimulate GLP-2 secretion or promote intestinal adaptation in SBS piglets, suggesting the necessity of intact intestinal tissue for efficacy [97].

#### 5.1.2. Bile Acid Replacement Therapy

Bile acid replacement therapy adjusts bile acid profiles to activate signaling pathways. Ursodeoxycholic acid (UDCA) is widely used to treat cholestatic hepatopathies due to its cytoprotective, anti-apoptotic, and immunomodulatory properties [98]. In SBS patients receiving PN, UDCA reduced hepatic cholesterol and triglyceride synthesis, preventing cholestasis and liver failure [99]. Dosages of 10–30 mg/kg/day in children and 10–15 mg/kg/day in adults improved biochemical markers of cholestasis within two months in IF patients undergoing total parental nutrition [100,101,102].

However, short-term improvement in biochemical parameters does not necessarily predict the outcome of IFALD patients, and UDCA may not be effective in patients with resected terminal ileum because of reduced UDCA absorption. It may indirectly stimulate FXR by favorably modulating bile acid profiles [103,104]. Notably, UDCA altered gut microbiota composition, decreasing *Bifidobacterium* and *Lactobacillus* abundance in liver- dysfunction patients, though the implications for IFALD remain unclear [105]. Taurine, a sulfur-containing amino acid, has also been investigated for its role in the development of IFALD in patients receiving PN. A reduction in plasma taurine levels may influence bile acid metabolism by decreasing the availability of tauro-conjugated bile acids. In a study by Schneider et al., patients with a jejunal length of less than 35 cm exhibited lower taurine levels compared to those with a jejunal length of 35 cm or more. However, total bile acid concentrations and the ratio of tauro-conjugated to glyco-conjugated bile acids did not significantly differ between the groups. Notably, long-term taurine supplementation (55 months) in patients with chronic cholestasis resulted in a significant reduction in aspartate aminotransferase levels, although markers of cholestasis remained unchanged. Synthetic or natural conjugated bile acids, along with taurine supplementation, warrant further investigation in IFALD settings [106,107,108,109,110].

### 5.2. Hormonal and Growth Factor Therapies

GLP-2 analogues have shown efficacy in reducing PN dependence in SBS-associated IF patients. Therefore, they show great potential for mitigating IFALD by enhancing enteral nutrition and reducing PN requirements. Yano et al. demonstrated that the use of GLP-2 at 1 μg/kg/h in SBS rat models on total PN significantly lower lobular inflammation score and NAFLD score, compared to placebo and to GLP-2 at 10 μg/kg/h [111]. However, the high-dose regimen, compared to the low-dose one, was more effective in mitigating intestinal mucosal atrophy. The authors attributed this discrepancy to factors such as the characteristics of the animal model, the experimental design involving massive bowel resection, and the young age of the rats (7 weeks). They concluded that the high dose of GLP-2 might have exceeded the hepatic processing capacity in their SBS rat model. Further studies are needed to determine the optimal GLP-2 dosage to promote intestinal adaptation while simultaneously mitigating hepatic steatosis in IFALD to assess its long-term efficacy in SBS patients, its efficacy after discontinuation, and potential complications.

In multiple studies, teduglutide, a GLP-2 analogue, reduced PN days and volumes [112,113,114,115,116]. It also improved stool frequency and consistency within 12 weeks [117]. Glepaglutide, a novel long-acting GLP-2 analogue, in a phase 2 clinical trial improved hepatic excretory function but increased liver stiffness through macrophages activation [118]. Similarly, apraglutide, another GLP-2 agonist, enhanced intestinal adaptation in SBS piglets [119]. Ongoing research on GLP-1/GLP-2 co-agonists in animal models shows potential benefits for gut morphometry and glucose metabolism, increased intestinal volume, and increased mucosal surface area [120].

Dipeptidyl peptidase-4 (DPP4) cleaves GLP-2 to its inactive form. Consistently, DPP4 inhibitors have demonstrated efficacy in reducing intrahepatic lipid levels in patients affected by type 2 diabetes and NAFLD [121]. In SBS animal models, DPP4 inhibitors improved intestinal mucosal capacity and adaptation, epithelial cell proliferation, glucose transport, and barrier function, while reducing bacterial translocation [122].

An interesting growth factor, which has recently been studied for the management of IFALD is the human hepatocyte growth factor (HGF). It is a hepatocyte mitogen with anti-inflammatory and antioxidant properties for the liver. Recombinant human hepatocyte growth factor (rh-HGF) in PN-fed SBS rats reduced hepatic steatosis, assessed by NAFLD activity score and inflammatory cell infiltration, increased FXR expression, lowered the expression of TLR-4 in the ileum and modulated gut microbiota composition [123]. Conversely, a multicenter trial on cholecystokinin-octapeptide (CCK-OP) revealed no significant impact on conjugated bilirubin levels or IFALD incidence [124].

Potential therapeutic strategies based on bile acids signaling or hormonal pathways are schematically summarized in Table 1.

### 5.3. Microbial Modulation

#### 5.3.1. Antibiotics

The use of empirical antibiotic therapy for the treatment of suspected small intestinal bacterial overgrowth (SIBO) or bacterial translocation and its potential to mitigate IFALD remains poorly investigated. Broad-spectrum antibiotics have been shown to ameliorate intestinal and hepatic inflammation and steatosis while decreasing small intestinal bacterial populations and enhancing anabolic responses in SBS animal models. However, they also reduce gut adaptation [126,127]. Given the role of anaerobic intestinal bacteria in IFALD pathogenesis, a retrospective study found that metronidazole administered during total parenteral nutrition was associated with the prevention of the expected increase in serum alkaline phosphatase, gamma-glutamyl transferase, and aspartate aminotransferase levels [128]. Similarly, another study demonstrated that metronidazole administration reduced or stabilized cholestatic enzyme levels in adult patients after 30 days of total parenteral nutrition [129]. However, despite the potential contribution of bacterial translocation to hepatocyte injury, prophylactic antibiotic use to prevent IFALD is not recommended. This is due to the risk of adverse antibiotic effects, the potential for developing bacterial resistance, insufficient available data, and limited evidence on long-term outcomes [130].

Erythromycin has also been investigated as a prokinetic agent to enhance enteral feeding in preterm infants with feeding intolerance [131]. High-dose oral erythromycin (12.5 mg/kg/dose every 6 h for 14 days) in preterm infants receiving PN reduced the duration of PN, decreased the incidence of IFALD, and significantly accelerated the time to full enteral nutrition [132]. Intermediate-dose erythromycin (5 mg/kg/dose every 6 h for 14 days) also decreased the incidence of IFALD and necrotizing enterocolitis by shortening PN duration and expediting full enteral nutrition [133]. Additionally, erythromycin demonstrated superior efficacy in facilitating full enteral feeding compared to UDCA [134].

#### 5.3.2. Probiotics

A higher number of studies focus instead on the effects of different probiotics on restoring gut and hepatic function in patients with SBS.

A retrospective study investigating probiotics for SIBO prevention in IF patients reported a higher prevalence of IFALD among non-probiotic users (54.4% vs. 39.1%, *p* = 0.005). However, multivariable analysis revealed that small bowel length (10–90 cm; odds ratio: 4.394, 95% confidence interval: 1.635–11.814; *p* = 0.003), and HPN use (OR: 4.502, 95% CI: 1.412–14.351; *p* = 0.011) rather than probiotic use (OR: 0.303–1.146, 95% CI: 0.303–1.146), significantly impacted IFALD risk [135]. However, different probiotic strains were analyzed in this study, thus potentially contributing to such inconsistent findings.

Indeed, not all microbial modifiers are able to balance gut adaptation and hepatic health when treating IFALD. Probiotics and prebiotics have shown improved gut adaptation and barrier function in SBS patients and animal models in most studies [136,137,138,139,140,141], though some others have failed to demonstrate significant benefits [142,143]. However, many probiotics used in these studies, such as *Lactobacillus johnsonii* and *Bifidobacterium breve*, encoded BSH genes—which increase bile acid deconjugation, reduce bile acid and lipid absorption—downregulate FXR signaling and stimulate hepatic bile acid and lipid synthesis. These effects may exacerbate bile acid and lipid dysmetabolism, promoting cholestasis and steatosis in IF patients [144].

Future research should prioritize careful selection of microbial modifiers for IF patients, with particular attention to their effects on bile acids and liver health. *Lactobacillus rhamnosus GG* (LGG) represents a promising candidate, having demonstrated benefits in gut adaptation, bacterial translocation reduction [137,145], and liver inflammation and fibrosis attenuation, thus preventing liver injury in cholestasis [146]. Although LGG does not directly convert bile acids, it enhances gut microbial BSH activity through compositional shifts. LGG also modulates FXR signaling: it downregulates FXR when bile acids are at normal level, while during cholestasis it upregulates FXR and suppresses hepatic bile acid synthesis via FXR-FGF15 signaling while enhancing bile acid excretion, potentially mitigating cholestasis-related liver injury [146]. Additionally, probiotics encoding bile acid inducible (bai) genes, such as *Clostridium scindens*, a bile-acid-7α-dehydroxylating intestinal bacterium which generates secondary bile acids, may offer therapeutic potential for IF patients lacking these critical metabolites [147].

Table 2 summarizes data about the role of probiotics in intestinal and hepatic outcomes from clinical or pre-clinical studies.

### 5.4. Surgical Treatments

#### 5.4.1. Surgical Lengthening Procedures

Small bowel length exceeding 38 cm, an intact ileocecal valve, intestinal continuity, preserved colon, and primary anastomosis are critical factors for survival and adaptation in SBS patients on PN [148]. Surgical lengthening procedures, such as longitudinal intestinal lengthening and tailoring (LILT), spiral intestinal lengthening and tailoring (SILT), and serial transverse enteroplasty (STEP), enhance absorptive surface area, decrease PN dependence, and facilitate enteral autonomy. These approaches have been shown to wean pediatric and adult patients from PN and normalize liver enzymes [149,150,151,152]. Notably, LILT improves peristalsis, nutrient absorption, and mucosal contact time, reducing bacterial overgrowth. A cohort study reported normalization of liver enzymes in patients with liver fibrosis following successful LILT and PN weaning [149].

#### 5.4.2. Transplantation

Intestinal transplantation is a life-saving intervention for IF patients with severe complications of PN, including advanced IFALD [153]. Two types of transplantation are utilized in IFALD management: isolated intestinal transplantation and combined liver-intestinal transplantation. Isolated intestinal transplantation is indicated for IFALD patients with mild to moderate liver disease, without portal hypertension or cirrhosis (platelet count >150,000/µL, minimal hepatosplenomegaly, total bilirubin < 6 mg/dL, stage 1–2 fibrosis on biopsy) [153,154]. In this scenario, key indications include recurrent dehydration despite intravenous fluids, frequent central line infections, thrombosis of two central venous channels, and high mortality risk due to underlying disease [154]. Combined liver–intestinal transplantation is reserved for patients with severe IFALD and portal hypertension, or end-stage liver disease. The liver and intestine are transplanted en bloc to avoid hilar dissection and minimize complications such as vascular and biliary issues [155].

Survival rates vary by age and transplant type. Pediatric intestinal recipients exhibit the highest survival rates (1-year: 86.2%; 5-year: 75.4%), whereas adult intestine-liver recipients have the lowest survival outcomes (1-year: 66.7%; 5-year: 42.6%) [156,157]. The immunogenicity of intestinal grafts, attributed to abundant lymphoid tissue and bacterial flora, predisposes recipients to infections, acute rejection, kidney disease, lymphoproliferative disorders, and graft-versus-host disease [158]. Advances in immunosuppressive protocols, including the incorporation of rituximab and bortezomib, belatacept, and bortezomib, or eculizumab and a C1 esterase inhibitor, together with further advancements in surgical techniques and post-operative care, hold promise for improving outcomes [158].

## 6. Conclusions

IFALD is a major complication of SBS, whose severity may progress to a life-threatening disease. For a long time, it has been considered mainly a consequence of PN, a crucial treatment for patients with IF. However, growing evidence demonstrates that mechanisms underlying IFALD are more complex and interconnected, encompassing factors other than PN, such as bile acid metabolism disruption, gut dysbiosis, and systemic inflammation. Such growing knowledge sheds new light on IFALD pathogenesis, helping physicians and researchers not only to select more appropriate PN formulations but also to identify further promising therapeutic targets.

## Figures and Tables

**Figure 1 biomolecules-15-00388-f001:**
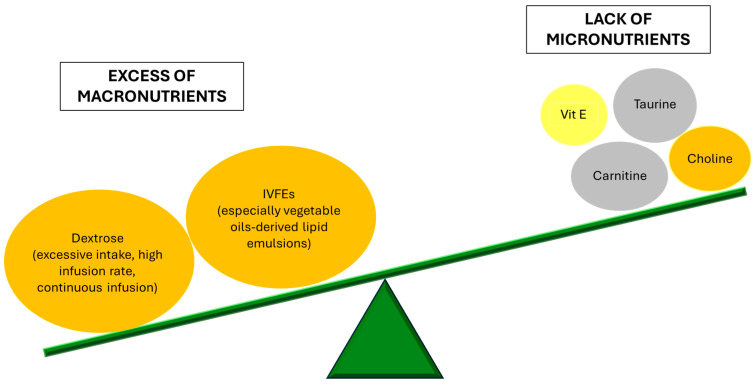
The imbalance in parenteral nutrition composition contributes to liver injury. Elements in orange have a confirmed role in promoting IFALD; the yellow color refers to elements with a role in preclinical studies, without any available data on humans; the grey color refers to elements with contrasting data from pre-clinical and clinical studies. See the main text for further details. IVFEs: intravenous fat emulsions.

**Figure 2 biomolecules-15-00388-f002:**
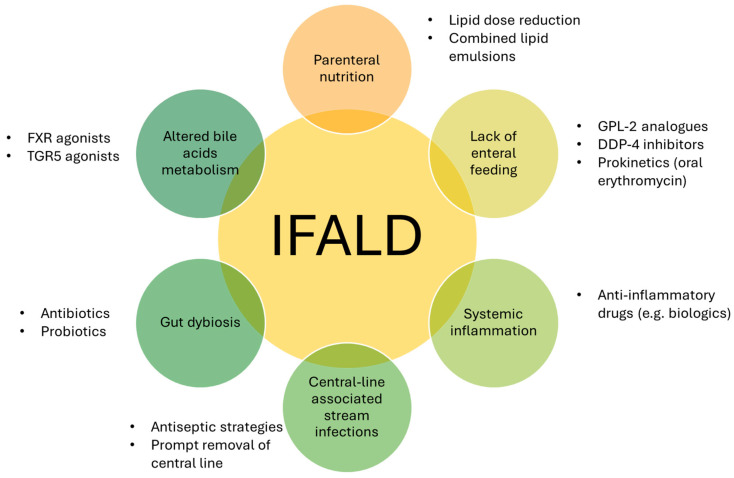
The main pathogenetic factors contributing to the complex pathogenesis of IFALD. FXR: farnesoid-X receptor; TGR5: Takeda-G-protein-coupled receptor 5; GLP-2: glucagon-like peptide-2; DPP4: dipeptidyl peptidase IV.

**Table 1 biomolecules-15-00388-t001:** Potential therapeutic strategies targeting bile acid or hormonal pathways.

Author, Year	Drug	Mechanism of Action	Study Model	Outcomes
Yi Cao et al., 2019 [93]	GW4064	FXR agonist	Animal model: SBR-ADL rats	Improved liver histology and serum liver enzymes, corrected BAs metabolism
Yang Liu et al., 2021 [94]	Tropifexor	FXR agonist	Animal model: neonatal piglet fed with PN	Prevented the increase of serum liver enzymes by increasing the abundance of intestinal bacteria producing bsh and CYP7A1 and altering the composition of BAs in serum, liver and intestinal content
Prue M Pereira-Fantini et al., 2017 [96]	OCA	FXR activation	Animal model: SBR piglets	Reduced fat malabsorption, but exacerbated liver histology
Gustavo Villalona et al., 2018 [95]	CDCA	FXR activation	Animal model: SBR piglets	Increased gut FXR, but not downstream FXR targets, not significant improvement in liver histology and cholestasis enzymes
Sen Lin et al., 2019 [97]	Ursolic acid	TGR5 agonist	Animal model: piglets receiving PN and SBR piglets	Increased GLP-2 secretion, but not intestinal adaptation after SBR. Liver outcomes not assessed.
Thomas Mouillot et al., 2020 [99]	UDCA	Indirect FXR activation	Human: SBS patients	Reduced hepatic cholesterol and triglucerides syntesis, decreased cholesterol and ALT serum concentrations
Spagnuolo M. et al., 1996 [100]	UDCA	Indirect FXR activation	Human: children on long-term TPN, with cholestasis	Normalization of biochemical markers of cholestasis within 4–8 weeks
De Marco G. et al., 2006 [125]	UDCA	Indirect FXR activation	Human: children on TPN, with IFALD	Decrease in serum GGT, ALT and direct bilirubin concentrations
Beau P. et al., 1994 [101]	UDCA	Indirect FXR activation	Human: SBS patients on TPN with cholestasis	Reduction of serum GGT and ALT, but not of ALP, bilirubin and AST.
Keisuke Yano et al., 2019 [111]	Different doses of GLP-2	GLP-2 increase	Animal model: SBR rats on TPN	Lower steatosis, lobular inflammation score and NAFLD score in the low-dose GLP-2 group
Rahim Mohammad Naimi et al., 2019 [118]	Glepaglutide	GLP-2 analogue	Human: SBS patients	Improved liver excretory function, but increased liver stiffness, probably due to activated resident liver macrophages
Ryo Sueyoshi et al., 2014 [122]	MK-0626	DPP4 inhibitor	Animal model: SBR mice	Improved intestinal adaptation. Liver outcomes not assessed
Keisuke Yano et al., 2022 [123]	rh-HGF	HGF agonist	Animal model: SBR rats on TPN	Reduced hepatic steatosis and inflammatory cell infiltration in the liver, higher FXR expression in the liver, lower TLR4 expression in the ileum, alteration of gut microbiota composition
Daniel H Teitelbaum et al., 2005 [124]	CCK-OP	CCK-OP increase	Human: neonates on TPN	No significant impact on conjugated bilirubin levels

FXR: Farnesoid-X-receptor; SBR: small bowel resection; ADL: associated liver disease; Bas: bile acids; PN: parental nutrition; bsh: bile salt hydrolase; CYP7A1: cholesterol 7α-hydroxylase; CDCA: chenodeoxycholic acid; OCA: obeticholic acid; UDCA: ursodeoxycholic acid; TGR5: Takeda G protein-coupled receptor 5; ALT: alanine aminotransferase; ALP: alkaline phosphatase, AST: aspartate aminotransferase, GGT: gamma-glutamyl transpeptidase, TPN: total parental nutrition; GLP-2: glucagon-like peptide-2; DPP4: dipeptidyl peptidase IV; rh-HGF: recombinant human hepatocyte growth factor; TLR: Toll-like receptor 4; CCK-OP: cholecystokinin-octapeptide.

**Table 2 biomolecules-15-00388-t002:** Role of probiotics in intestinal and hepatic function in patients with IFALD.

Author, Year	Probiotic	Study Model	Intestinal Outcomes	Liver Outcomes
Mohammad Alomari et al., 2020 [135]	Unspecified probiotics	Human: patients with IF on HPN or HIVF	Not applied	Lower prevalence of IFALD in probiotic users, but not significant at the multivariate analysis
Jiang Wu et al., 2018 [137]	*Lactobacillus rhamnosus GG*	Animal model: SBR rats	Reduced bacterial translocation and intestinal permeability, lower levels of serum endotoxin and tumor necrotizing factor alpha in ileum	Not applied
Hannah G Piper et al., 2020 [142]	*Lactobacillus rhamnosus* and *Lactobacillus johnsonii*	Human: SBS children	No significant differences	Not applied
Timothy A Sentongo et al., 2008 [143]	*Lactobacillus rhamnosus GG*	Human: SBS children	No significant effect on intestinal permeability	Not applied
Jorge G Mogilner et al., 2007 [145]	*Lactobacillus rhamnosus GG*	Animal model: SBR rats	Reduced bacterial translocation, decrease enterocytes apoptosis and increased crypt-depth in ileum	Not applied
Keiichi Uchida et al., 2007 [138]	*Bifidobacterium breve, Lactobacillus casei* and galactooligosaccharides	Human: SBS children	Improved intestinal adaptation, potsitive gut microbiota modulation and increased SCFA levels in the feces. Increased serum pre-albumin levels and improved systemic immunonutritional status	Not applied
I Eizaguirre et al., 2002 [139]	*Bifidobacterium lactis*	Animal model: SBR rats	Reduced incidence of bacterial translocation	Not applied
Ethan A Mezoff et al., 2016 [140]	Human milk oligosaccharide 2′-fucosyllactose	Animal model: ICR mice	Increased weight gain and crypth depth. Increased energy availability through gut microbial modulation	Not applied
Yutaka Kanamori et al., 2004 [141]	*Bifidobacterium breve, Lactobacillus casei,* and galactooligosaccharides	Human: SBS patients	Accelerated body weight gain, positive modulation of gut microbiota, increased SCFA in the feces	Not applied

IF: intestinal failure; SBS: short bowel syndrome; HPN: home parenteral nutrition; HIVFs: home intravenous fluids; ICR: ileocecal resection; SCFA: short-chain fatty acids.

## Data Availability

No new data were created or analyzed in this study. Data sharing is not applicable to this article.

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
