# Peer review of "Intestinal-Failure-Associated Liver Disease: Beyond Parenteral Nutrition"

_biomolecules, 2025, doi:10.3390/biom15030388_

Round 1

Reviewer 1 Report

Comments and Suggestions for Authors The only comment I have is regarding the sentence: "Patients receiving PN have a lack of micronutrients such as carnitine, taurine, choline, and vitamins C and E." This needs to be corrected, as vitamins C and E are actually included in intravenous nutritional mixtures (lines 160-161). Overall, this article thoughtfully examines various aspects of liver disease related to gastrointestinal failure. I have no other critical comments to offer aside from the correction mentioned above.

Author Response

The only comment I have is regarding the sentence: "Patients receiving PN have a lack of micronutrients such as carnitine, taurine, choline, and vitamins C and E." This needs to be corrected, as vitamins C and E are actually included in intravenous nutritional mixtures (lines 160-161). Overall, this article thoughtfully examines various aspects of liver disease related to gastrointestinal failure. I have no other critical comments to offer aside from the correction mentioned above.

We thank the reviewer for this encouraging comment. We have modified the text as suggested.

Reviewer 2 Report

Comments and Suggestions for Authors

Overall, a well-written, clear review of the current literature on factors leading to the development of IFALD outside of the role of parenteral lipid forumlations.

General Comments:

Given the large number of abbreviations, if the journal allows it, a list of all the abbreviations would be helpful to make reading it easier.

In section 2, you mention some types of blood based measurement indices that have been validated in other liver disease may not work in the assessment of IFALD. Perhaps adding one – two sentences to explain that reason would be appropriate.

In section 3, line 206, you state that the above data highlights the importance of avoiding short-duration IVFG infusions. However, the data you are referencing is against the use of high infusion rates, not duration. Please clarify.

Line 210, I don’t know how accurate it would be to state that most of the commercially available IVFEs in the US currently use soybean oil or safflower oil. Soybean oil, olive oil, mcts, and fish oil lipid emulsions are all more prevalent than safflower oil. I would omit safflower from this statement. Or if the point is about high omega-6 fatty acids, you should state that the main sources is primarily soybean oil and that historically safflower oil was used.

While Section 3 presents studies favoring SMOFlipid over Intralipid, a more rigorous analysis requires the inclusion of single-center studies that have found no significant benefit. Addressing these conflicting findings is crucial for a comprehensive understanding of SMOFlipid's efficacy.

Figure 1 presents a helpful overview of factors contributing to liver injury. However, I have concerns regarding the accuracy of the right side of the figure. While the left side accurately depicts established factors supported by both basic and clinical findings, the right side includes components absent from IV formulations and depicts their roles in disease development without clear clinical confirmation. To enhance clarity and avoid overstating their established roles, I suggest using distinct color coding and a legend to differentiate between factors with confirmed clinical evidence and those with potential or hypothesized links.

In the section on bile acid receptor ligands, there is a brief mention of TGR5. This receptor was not introduced in prior sections as a regulator of bile acid homeostasis, so a little more detail on the receptor in this section would be appropriate to highlight its significance. Also, a mention that it responds to different bile acids than FXR would be appropriate to highlight the functional importance of bile acids in regulating these receptors.

Line 366-367. While taurine is mentioned briefly in a preceding section, this sentence seems to come out of nowhere as there is no mention to the relevance of conjugated bile acids in activating bile acid receptors or having therapeutic benefits in the preceding paragraphs. Some information about this leading into the statement are needed to give relevance to it.

Line 372: You bring up dosing of GLP2 in a rat study and give information that shows the low dosing of GLP2 is more effective than the high dosing. Given this does not follow normal expectations of the effectiveness of higher dosing of GLP2 on gut growth, could you briefly address this uncommon disparity.

Specific Edits:

Line 11: I don’t think I understand the use of the word “exit” in this context.

Line 14: Or you trying to say it ranges from “steatosis to cholestasis” and “fibrosis to end-stage liver disease”. As it is worded the use of the term “ranges” makes the sentence a little confusing.

Line 15: include the word “been” between “has” and “largely”

Line 44: Use of the word “therefore” in this transition is a little awkward. Possibly another word or phrasing to define the region of intestine remining from this classification of SBS

Line 69: Changes “several” to “multiple lines of evidence” or some other phrasing for clarity.  

Line 105: “formally” to “formal”

Line 114: “mild” to “mildly”

Line 136: fix “though none of,” to “though none of the patients”.

Line 176: you don’t need to capitalize “very low density lipoprotein”

Line 270: remove the word “the”

Line 291: not sure what the term “exit” is meant to mean.

Line 300: change “furtherly” to “further”

Line 302: “an” to “a”

Line 315: should define LPS is this is the first use and but the word “a” after the comma.

Line 317: Do you meal TLR ligands other than LPS. If so, state the ligands. Or make the connection that LPS is a ligand for TLRs.

Line 436: This possibly would make more sense as a transition sentence at the end of the previous section Line 434 rather than the introduction to the new section.

Line 444: change “thus probably explaining” to “potentially contributing to”

Line 446: change to “have shown improved”

Author Response

Overall, a well-written, clear review of the current literature on factors leading to the development of IFALD outside of the role of parenteral lipid formulations.

General Comments:

Given the large number of abbreviations, if the journal allows it, a list of all the abbreviations would be helpful to make reading it easier. Thanks for the suggestion. We have added a table summarizing abbreviations at the end of the main text.

In section 2, you mention some types of blood based measurement indices that have been validated in other liver disease may not work in the assessment of IFALD. Perhaps adding one – two sentences to explain that reason would be appropriate. According to the reviewer’s suggestion, we have added some sentences to explain the possible mechanisms underlying such differences.

In section 3, line 206, you state that the above data highlights the importance of avoiding short-duration IVFG infusions. However, the data you are referencing is against the use of high infusion rates, not duration. Please clarify. We tried to rewrite the paragraph in order to better clarify this aspect.

Line 210, I don’t know how accurate it would be to state that most of the commercially available IVFEs in the US currently use soybean oil or safflower oil. Soybean oil, olive oil, mcts, and fish oil lipid emulsions are all more prevalent than safflower oil. I would omit safflower from this statement. Or if the point is about high omega-6 fatty acids, you should state that the main sources is primarily soybean oil and that historically safflower oil was used. We have modified the sentence according to reviewer’s suggestion.

While Section 3 presents studies favoring SMOFlipid over Intralipid, a more rigorous analysis requires the inclusion of single-center studies that have found no significant benefit. Addressing these conflicting findings is crucial for a comprehensive understanding of SMOFlipid's efficacy. Thanks for the comment. We have further reviewed the literature and added data from some recent studies. As far as we know, we have summarized the most updated evidence on the topic.

Figure 1 presents a helpful overview of factors contributing to liver injury. However, I have concerns regarding the accuracy of the right side of the figure. While the left side accurately depicts established factors supported by both basic and clinical findings, the right side includes components absent from IV formulations and depicts their roles in disease development without clear clinical confirmation. To enhance clarity and avoid overstating their established roles, I suggest using distinct color coding and a legend to differentiate between factors with confirmed clinical evidence and those with potential or hypothesized links. We thank the reviewer for this suggestion. We have introduced different colors according to reviewer’s comment and we have clarified more details in the figure legend.

In the section on bile acid receptor ligands, there is a brief mention of TGR5. This receptor was not introduced in prior sections as a regulator of bile acid homeostasis, so a little more detail on the receptor in this section would be appropriate to highlight its significance. Also, a mention that it responds to different bile acids than FXR would be appropriate to highlight the functional importance of bile acids in regulating these receptors. According to reviewer’s comment, we have implemented the text providing more details on TGR5 and FXR.

Line 366-367. While taurine is mentioned briefly in a preceding section, this sentence seems to come out of nowhere as there is no mention to the relevance of conjugated bile acids in activating bile acid receptors or having therapeutic benefits in the preceding paragraphs. Some information about this leading into the statement are needed to give relevance to it. Thanks for the suggestion. We have added connecting sentences and some more information about taurine to make the paragraph easier to understand.

Line 372: You bring up dosing of GLP2 in a rat study and give information that shows the low dosing of GLP2 is more effective than the high dosing. Given this does not follow normal expectations of the effectiveness of higher dosing of GLP2 on gut growth, could you briefly address this uncommon disparity. We have modified the paragraph to better explain this point.

Specific Edits:

  • Line 11: I don’t think I understand the use of the word “exit” in this context. R: we change the verb with “lead to” to better clarify.
  • Line 14: Or you trying to say it ranges from “steatosis to cholestasis” and “fibrosis to end-stage liver disease”. As it is worded the use of the term “ranges” makes the sentence a little confusing. R: the sentence has been corrected
  • Line 15: include the word “been” between “has” and “largely”. R: the sentence has been corrected
  • Line 44: Use of the word “therefore” in this transition is a little awkward. Possibly another word or phrasing to define the region of intestine remining from this classification of SBS. R: the sentence has been corrected
  • Line 69: Changes “several” to “multiple lines of evidence” or some other phrasing for clarity.  R: the sentence has been corrected
  • Line 105: “formally” to “formal” R: the sentence has been corrected
  • Line 114: “mild” to “mildly” R: the sentence has been corrected
  • Line 136: fix “though none of,” to “though none of the patients”. R: the sentence has been corrected
  • Line 176: you don’t need to capitalize “very low density lipoprotein” R: the sentence has been corrected
  • Line 270: remove the word “the” R: the sentence has been corrected
  • Line 291: not sure what the term “exit” is meant to mean. R: we have changed with the verb “contribute to” to clarify the meaning.
  • Line 300: change “furtherly” to “further” R: the sentence has been corrected
  • Line 302: “an” to “a” R: the sentence has been corrected
  • Line 315: should define LPS is this is the first use and but the word “a” after the comma. R: We added the word “a” after the comma. The abbreviation LPS is defined in line 87, when it appears in the text for the first time
  • Line 317: Do you meal TLR ligands other than LPS. If so, state the ligands. Or make the connection that LPS is a ligand for TLRs. We have specified LPS is a ligand for TLRs.
  • Line 436: This possibly would make more sense as a transition sentence at the end of the previous section Line 434 rather than the introduction to the new section. R: thanks for the comment. However, we chose to place that sentence in the new paragraph to better differentiate antibiotics and probiotics-based treatments.
  • Line 444: change “thus probably explaining” to “potentially contributing to” R: the sentence has been corrected
  • Line 446: change to “have shown improved” R: the sentence has been corrected